# Chimeric antigen receptors that trigger phagocytosis

**Meghan A Morrissey[1,2†], Adam P Williamson[1,2†], Adriana M Steinbach[1,2], Edward W Roberts[3], Nadja Kern[1,2], Mark B Headley[3], Ronald D Vale[1,2]***

[1]Department of Cellular and Molecular Pharmacology, University of California, San Francisco, San Francisco, United States; [2]Howard Hughes Medical Institute, University of California, San Francisco, San Francisco, United States; [3]Department of Pathology, University of California, San Francisco, San Francisco, United States

**Abstract** Chimeric antigen receptors (CARs) are synthetic receptors that reprogram T cells to kill cancer. The success of CAR-T cell therapies highlights the promise of programmed immunity and suggests that applying CAR strategies to other immune cell lineages may be beneficial. Here, we engineered a family of Chimeric Antigen Receptors for Phagocytosis (CAR-Ps) that direct macrophages to engulf specific targets, including cancer cells. CAR-Ps consist of an extracellular antibody fragment, which can be modified to direct CAR-P activity towards specific antigens. By screening a panel of engulfment receptor intracellular domains, we found that the cytosolic domains from Megf10 and FcRɣ robustly triggered engulfment independently of their native extracellular domain. We show that CAR-Ps drive specific engulfment of antigen-coated synthetic particles and whole human cancer cells. Addition of a tandem PI3K recruitment domain increased cancer cell engulfment. Finally, we show that CAR-P expressing murine macrophages reduce cancer cell number in co-culture by over 40%.

DOI: https://doi.org/10.7554/eLife.36688.001

**\*For correspondence:**
Ron.Vale@ucsf.edu

[†]These authors contributed equally to this work

**Competing interests:** The authors declare that no competing interests exist.

## Introduction

Chimeric antigen receptors (CARs) are synthetic transmembrane receptors that redirect T cell activity towards clinically relevant targets (reviewed in [*Lim et al., 2017*; *Fesnak et al., 2016*]). The CAR-T receptor contains an extracellular single chain antibody fragment (scFv) that recognizes known tumor antigens, and intracellular signaling domains from the T Cell Receptor (TCR) and costimulatory molecules that trigger T cell activation (*Fesnak et al., 2016*; *Kochenderfer et al., 2009*). CAR-T cells recognizing CD19, a marker expressed at high levels on the surface of B cells and B cell-derived malignancies, have been used successfully to target hematological malignancies with 70–90% of patients showing measurable improvement (*Lim et al., 2017*; *Engel et al., 1995*; *Haso et al., 2013*). The success of CAR-T suggests that programming immune cells to target cancer might be a broadly applicable approach.

Macrophages are critical effectors of the innate immune system, responsible for engulfing debris and pathogens. Harnessing macrophages to combat tumor growth is of longstanding interest (*Alvey and Discher, 2017*; *Lee et al., 2016*). Macrophages are uniquely capable of penetrating solid tumors, while other immune cells, like T cells, are physically excluded or inactivated (*Lim et al., 2017*; *Lee et al., 2016*). This suggests that engineered macrophages may augment existing T cell-based therapies. Early efforts transferring healthy macrophages into cancer patients failed to inhibit tumor growth, suggesting that macrophages require additional signals to direct their activity towards tumors (*Lacerna et al., 1988*; *Andreesen et al., 1990*). Antibody blockade of CD47, a negative regulator of phagocytosis, reduced tumor burden, indicating that shifting the balance in favor of macrophage activation and engulfment is a promising therapeutic avenue (*Majeti et al., 2009*;

**eLife digest** Our immune system constantly patrols our body, looking to eliminate cancerous cells and harmful microbes. It can spot these threats because it recognizes certain signals at the surface of dangerous cells. However, cancer cells often find ways to 'hide' from our immune system.

Chimeric antigen receptors, or CARs, are receptors designed in a laboratory to attach to specific proteins that are found on a cancer cell. These receptors tell immune cells, such as T cells, to attack cancers. T cells that carry CARs are already used to treat people with blood cancers. Yet, these immune cells are not good at penetrating a solid tumor to kill the cells inside, which limits their use.

Macrophages are a group of immune cells that can make their way inside tumors and travel to cancers that the rest of the immune system cannot reach. They defend our body by 'swallowing' harmful cells. Would it then be possible to use CARs to program macrophages to 'eat' cancer cells?

Morrissey, Williamson et al. created a new type of CARs, named CAR-P, and introduced it in macrophages. These cells were then able to recognize and attack beads covered in proteins found on cancer cells. The modified macrophages could also limit the growth of live cancer cells in a dish by 'biting' and even 'eating' them. While these results are promising in the laboratory, the next step is to test whether these reprogrammed macrophages can recognize and fight cancers in living animals.

DOI: https://doi.org/10.7554/eLife.36688.002

*Chao et al., 2010*; *Jaiswal et al., 2009*; *Tseng et al., 2013*). Here, we report a family of chimeric antigen receptors that activate phagocytosis of cancer cells based on recognition of defined cell surface markers, resulting in significantly reduced cancer cell growth.

## Results

To program engulfment towards a target antigen, we created a CAR strategy using the CAR-T design as a guide (*Fesnak et al., 2016*). We call this new class of synthetic receptors Chimeric Antigen Receptors for Phagocytosis (CAR-Ps). The CAR-P molecules contain the extracellular single-chain antibody variable fragment (scFv) recognizing the B cell antigen CD19 ($\alpha$CD19) and the CD8 transmembrane domain present in the $\alpha$CD19 CAR-T (*Fesnak et al., 2016*; *Kochenderfer et al., 2009*). To identify cytoplasmic domains capable of promoting phagocytosis, we screened a library of known murine phagocytic receptors: Megf10 (*Figure 1a*), the common $\gamma$ subunit of Fc receptors (FcR$\gamma$), Bai1, and MerTK (*Penberthy and Ravichandran, 2016*). FcR triggers engulfment of antibody-bound particles, while the other receptors recognize apoptotic corpses (*Freeman and Grinstein, 2014*; *Penberthy and Ravichandran, 2016*). We also made a receptor containing an extracellular $\alpha$CD19 antibody fragment and a cytoplasmic GFP, but no signaling domain, to test whether adhesion mediated by the $\alpha$CD19 antibody fragment is sufficient to induce engulfment (*Figure 1a*; CAR-P$^{GFP}$).

To assay our library of CAR-Ps, we introduced each CAR-P into J774A.1 murine macrophages by lentiviral infection. As an engulfment target, we used 5 $\mu$m diameter silica beads coated with a supported lipid bilayer. A His$_8$-tagged extracellular domain of CD19 was bound to a NiNTA-lipid incorporated into the supported lipid bilayers. Macrophages expressing a CAR-P with the Megf10 (CAR-P$^{Megf10}$) or FcR$\gamma$ (CAR-P$^{FcR\gamma}$) intracellular domain promoted significant engulfment of CD19 beads compared to macrophages with no CAR (*Figure 1b,c*, *Figure 1—video 1*). Macrophages expressing CAR-P$^{Bai1}$, CAR-P$^{MerTK}$, and the adhesion-only CAR-P$^{GFP}$ did not bind the CD19 beads even though these CAR-Ps are present at the cell surface (*Figure 1b,c*, *Figure 1—figure supplement 1*). To confirm that the CAR-P was a viable strategy for redirecting primary macrophages, we expressed the CAR-P$^{FcR\gamma}$ in primary murine bone marrow derived macrophages and found that these transfected primary cells also were able to trigger engulfment of CD19 beads (*Figure 1d*).

Next we asked if the CAR-P strategy could target a different antigen. Because CAR-P$^{Megf10}$ performed well in our initial screen (*Figure 1a*), we developed $\alpha$CD22 CAR-P$^{Megf10}$ using a previously developed $\alpha$CD22 antibody fragment (*Xiao et al., 2009*; *Haso et al., 2013*). Consistent with our results using $\alpha$CD19-based CARs, $\alpha$CD22 CAR-P$^{Megf10}$ promoted engulfment of CD22 beads (*Figure 2a*). To confirm antigen specificity of CAR-P, we incubated $\alpha$CD19 CAR-P$^{Megf10}$

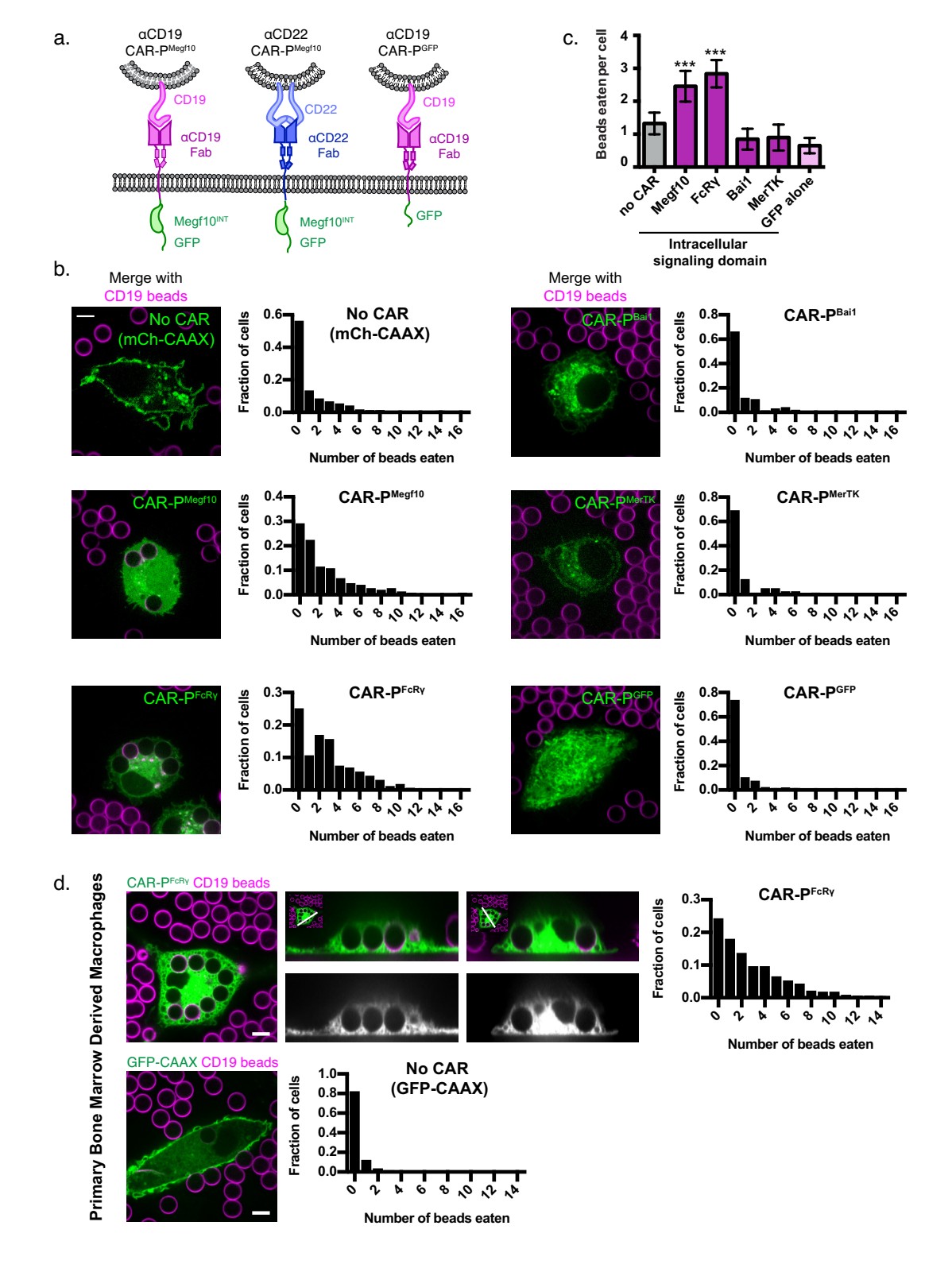

**Figure 1.** Identification of intracellular signaling region for CAR-P. (**A**) Schematics show the structure of CAR-P constructs. An αCD19 (purple) or αCD22 (blue, center) scFv directs CAR specificity. Intracellular signaling domains from Megf10 or the indicated engulfment receptor (green) activate engulfment. CAR-P^GFP contains only GFP and no intracellular signaling domains (right). All constructs include a transmembrane domain from CD8 and a C-terminal GFP. (**B**) J774A.1 macrophages expressing αCD19 CAR-P with the indicated intracellular signaling domain (green) engulf 5 μm silica beads

*Figure 1 continued on next page*

*Figure 1 continued*

covered with a supported lipid bilayer containing His-tagged CD19 extracellular domain. The beads were visualized with atto390-labeled lipid incorporated into the supported lipid bilayer (magenta). Cells infected with the cell membrane marker, mCherry-CAAX, were used as a control (no CAR, top left). To the right of each image is a histogram depicting the frequency of cells engulfing the indicated number of beads. The average number of beads eaten per cell is quantified in (C). (D) Bone marrow derived macrophages were infected with CAR-P$^{FcR\gamma}$ or GFP-CAAX (green, left and center top; grey, center bottom) and incubated with CD19 beads (magenta) for 45 min. Images show an x-y plane through the center of the engulfed beads (left), or a cross section (center) of the z plane indicated in the inset panel (white line). The histogram depicts the number of cells engulfing the indicated number of beads. The scale bar indicates 5 µm and n = 78–163 cells per condition, collected during three separate experiments. Error bars denote 95% confidence intervals and *** indicates p<0.0001 compared to mCherry-CAAX control by Kruskal-Wallis test with Dunn's multiple comparison correction.

DOI: https://doi.org/10.7554/eLife.36688.003

The following video and figure supplement are available for figure 1:

**Figure supplement 1.** Expression level of CAR-P constructs in macrophages.

DOI: https://doi.org/10.7554/eLife.36688.004

**Figure 1—video 1.** CAR-P$^{Megf10}$ macrophage engulfs silica beads

DOI: https://doi.org/10.7554/eLife.36688.005

macrophages with CD22 beads, and αCD22 CAR-P$^{Megf10}$ macrophages with CD19 beads. CD19 beads were not eaten by αCD22 CAR-P$^{Megf10}$ macrophages, and CD22 beads were not eaten by αCD19 CAR-P$^{Megf10}$ macrophages (*Figure 2a*). These data indicate that CAR-P$^{Megf10}$ specifically

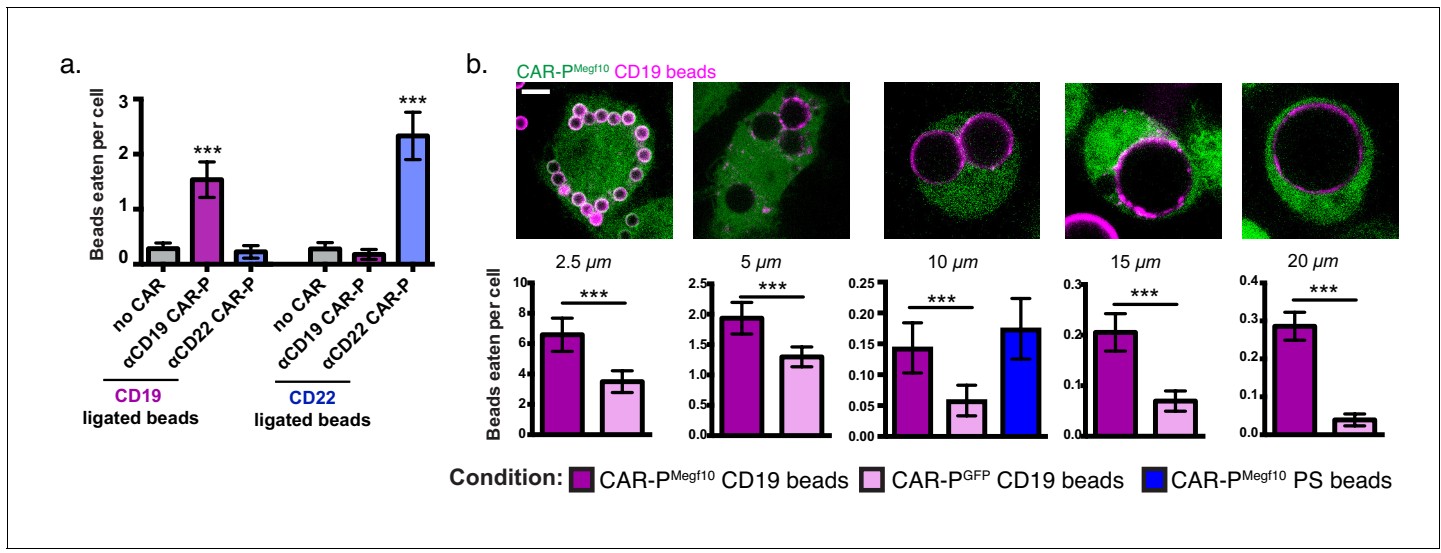

**Figure 2.** CAR-P expression drives specific engulfment of diverse beads. (**A**) Macrophages infected with the αCD19 (purple) or αCD22 (blue) CAR-P$^{Megf10}$ or mCherry-CAAX control were fed 5 µm beads ligated with either CD19 (left) or CD22 (right). Engulfment is quantified as the mean beads eaten per cell. The fraction of phagocytic cells is as follows: 31/144 GFP-CAAX cells engulfed CD19 beads, 87/149 αCD19 CAR-P$^{Megf10}$ engulfed CD19 beads, 20/142 αCD22 CAR-P$^{Megf10}$ engulfed CD19 beads, 28/140 GFP-CAAX cells engulfed CD22 beads, 18/151 αCD19 CAR-P$^{Megf10}$ engulfed CD22 beads, 103/148 αCD22 CAR-P$^{Megf10}$ engulfed CD22 beads (pooled data was collected during three separate experiments). Error bars denote 95% confidence intervals and *** indicates p<0.0001 compared to mCherry-CAAX control by Kruskal-Wallis test with Dunn's multiple comparison correction. (**B**) J774A.1 macrophages expressing the αCD19 CAR-P$^{Megf10}$ (green) were fed beads of various sizes (magenta, diameter of bead indicated below image). The beads were covered in a supported lipid bilayer ligated to His-tagged CD19 extracellular domain and the number of beads engulfed per cell is reported below each image (magenta bars indicate CAR-P$^{Megf10}$ macrophages and pink bars indicate CAR-P$^{GFP}$). The αCD19 CAR-P$^{Megf10}$ macrophages were also incubated with 10 µm beads coated in phosphatidylserine (PS) and ICAM-1 (blue bar in graph, 51/390 cells engulfed a bead). The fraction of cells engulfing a CD19 bead is as follows: 135/169 CAR-P$^{Megf10}$ and 134/187 CAR-P$^{GFP}$ cells engulfed 2.5 um bead, 126/395 CAR-P$^{Megf10}$ and 112/499 CAR-P$^{GFP}$ cells engulfed a 5 um bead, 48/377 CAR-P$^{Megf10}$ and 21/378 CAR-P$^{GFP}$ cells engulfed a 10 um bead, 120/706 CAR-P$^{Megf10}$ and 45/675 CAR-P$^{GFP}$ cells engulfed a 15 um bead, 194/760 CAR-P$^{Megf10}$ and 23/587 CAR-P$^{GFP}$ cells engulfed a 20 um bead (data is pooled from at least three separate experiments). Error bars denote 95% confidence intervals of the mean. *** indicates p<0.0001 respectively by Mann-Whitney test. All scale bars represent 5 µm.

DOI: https://doi.org/10.7554/eLife.36688.006

triggers engulfment in response to the target ligand and that the CAR-P strategy is able to target multiple cancer antigens.

To further define the capabilities of the CAR-P, we assessed the capacity of CAR-P-expressing macrophages to engulf variably sized targets. We found that CAR-P$^{Megf10}$ was able to trigger specific engulfment of beads ranging from 2.5 μm to 20 μm in diameter, with higher specificity above background engulfment being demonstrated for the larger beads (*Figure 2b*). The high background in this assay is due to heterogeneity in the bilayers on beads purchased from a different manufacturer (Corpuscular) than previous assays. For the 10 μm bead condition, we also tested the phagocytic efficiency of beads containing the endogenous Megf10 ligand, phosphatidylserine. We found that CAR-P$^{Megf10}$ macrophages engulfed CD19 beads and beads containing 10% phosphatidylserine and the adhesion molecule ICAM-1 at a similar frequency (*Figure 2b*). This indicates that the CAR-P is comparably efficient to the endogenous system.

To determine if the CAR-P$^{Megf10}$ initiates active signaling at the synapse between the macrophage and target, we stained for phosphotyrosine. Macrophages expressing CAR-P$^{Megf10}$ exhibited an increase in phosphotyrosine at the synapse, while macrophages expressing CAR-P$^{GFP}$ did not show this enrichment (*Figure 3a*). Consistent with previous reports, we found that F-actin also was enriched at the cell bead synapse (*Figure 3—figure supplement 1*). This result suggests that CAR-P$^{Megf10}$ initiates engulfment through a localized signaling cascade involving tyrosine phosphorylation.

Both successful CAR-P intracellular domains (from FcRγ and Megf10) have cytosolic Immunoreceptor Tyrosine-based Activation Motifs (ITAMs) that are phosphorylated by Src family kinases. Based on this observation, we hypothesized that the expression of an alternate ITAM-containing receptor might initiate phagocytosis when expressed in macrophages. The CD3ζ subunit of the T cell receptor contains three ITAM motifs. To test if the CD3ζ chain was able to activate phagocytic signaling, we transduced macrophages with the first generation CAR-T (*Figure 3b*). The CAR-T was able to trigger engulfment of CD19 beads to a comparable extent as CAR-P$^{Megf10}$ (*Figure 3c*). In T cells, phosphorylated ITAMs in CD3ζ bind to tandem SH2 domains (tSH2) in the kinase ZAP70. Zap70 is not expressed in macrophages, but Syk, a phagocytic signaling effector and tSH2 domain containing protein, is expressed at high levels (*Andreu et al., 2017*). Previous work suggested that Syk kinase can also bind to the CD3ζ ITAMs (*Bu et al., 1995*), indicating that the CAR-T may promote engulfment through a similar mechanism as CAR-P$^{FcRγ}$. To quantitatively compare the interaction between Syk$^{tSH2}$ and CD3ζ or FcRγ in a membrane proximal system recapitulating physiological geometry, we a used liposome-based assay (*Figure 3d* [*Hui and Vale, 2014*]). In this system, His$_{10}$-CD3ζ and His$_{10}$-Lck (the kinase that phosphorylates CD3ζ) are bound to a liposome via NiNTA-lipids and the binding of labeled tandem SH2 domains to phosphorylated CD3ζ was measured using fluorescence quenching. Our results show that Syk-tSH2 binds the CD3ζ and FcRγ with comparable affinity (~15 nM and ~30 nM respectively, *Figure 3d*). Collectively, these results demonstrate that the TCR CD3ζ chain can promote phagocytosis in a CAR-P, likely through the recruitment of Syk kinase.

We next sought to program engulfment towards a cellular target. We incubated the CAR-P$^{Megf10}$ and CAR-P$^{FcRγ}$ macrophages with cancerous Raji B cells that express high levels of endogenous CD19. Strikingly, the majority of CAR-P-expressing macrophages internalized bites of the target cell (*Figure 4a*, *Figure 4—video 1*, 78% of CAR-P$^{Megf10}$ and 85% of CAR-P$^{FcRγ}$ macrophages internalized bites within 90 min). The biting phenotype resembles trogocytosis, or nibbling of live cells, which has been reported previously in immune cells (*Joly and Hudrisier, 2003*). This process was dependent on the ITAM-bearing intracellular signaling domain, as removing the signaling domain (CAR-P$^{GFP}$) dramatically reduced trogocytosis (*Figure 4a*). Enrichment of phosphotyrosine at the cell-cell synapse further supports active signaling initiating trogocytosis (*Figure 4—figure supplement 1*). The CAR-P module also was able to induce trogocytosis in non-professional phagocytes, human NIH 3T3 fibroblast cells (*Figure 4—figure supplement 2*). This suggests that the CAR-P can promote cancer antigen-dependent engulfment by both professional and non-professional phagocytes.

We next focused on engineering strategies to engulf whole human cancer cells. We observed that macrophages expressing the CAR-P$^{Megf10}$ or CAR-P$^{FcRγ}$ were capable of engulfing whole Raji B cells (2 cancer cells eaten per 100 macrophages in a 4–8 hr window for both CAR-P$^{Megf10}$ or CAR-P$^{FcRγ}$, *Figure 4b,e*, *Figure 4—video 2*). Whole cell engulfment was infrequent but trogocytosis was

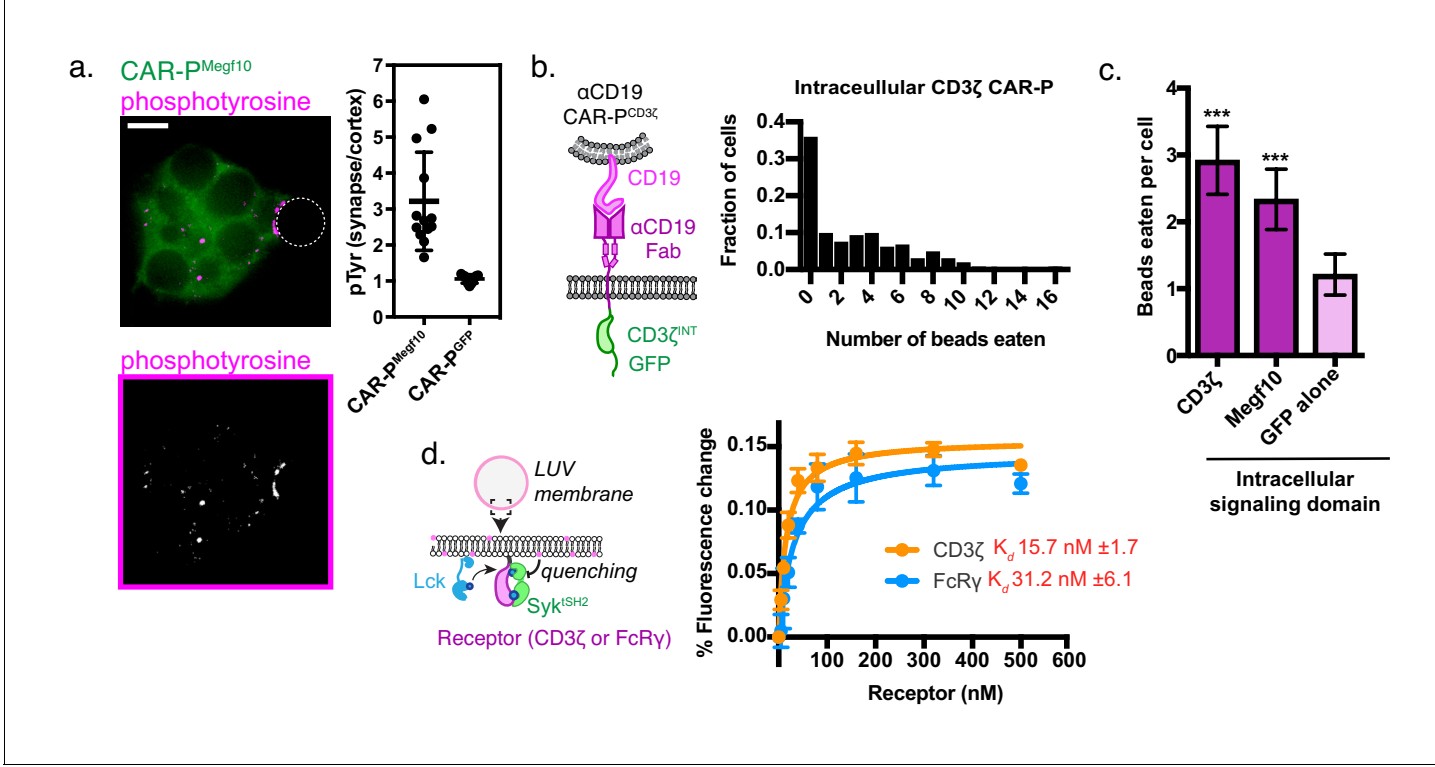

**Figure 3.** A phosphorylated ITAM at the cell-target synapse drives engulfment. (A) Macrophages expressing αCD19 CAR-P^Megf10 (green, top) or αCD19 CAR-P^GFP were incubated with CD19-ligated beads (position indicated with dotted line), fixed and stained for phosphotyrosine (magenta, top; greyscale, bottom). The fold enrichment of phosphotyrosine at the cell-bead synapse compared to the cell cortex is graphed on the right (n ≥ 11; each dot represents one cell-bead synapse; lines represent the mean ±one standard deviation). (B) Schematic shows the structure of CAR-P constructs in the plot at right. An αCD19 (purple) scFv directs CAR specificity. The intracellular signaling domains from CD3ζ activate engulfment. On the right is a histogram depicting the fraction of macrophages engulfing the indicated number of CD19-coated beads. (C) Comparison showing the average number of beads eaten per cell in J774A.1 macrophages expressing αCD19 CAR-Ps with the indicated intracellular signaling domain. 5 µm silica beads covered with a supported lipid bilayer containing His-tagged CD19 extracellular domain were used as an engulfment target (n = 156–167 cells per condition collected during three separate experiments). Error bars denote 95% confidence intervals and *** indicates p<0.0001 compared to CAR-P^GFP control by Kruskal-Wallis test with Dunn's multiple comparison in correction. (D) Model of the liposome-based fluorescence quenching assay used to determine affinity between the Syk tSH2 domains and the receptor tails of CD3ζ and FcRγ, two intracellular signaling domains that promote engulfment. Binding between the Syk tSH2 reporter (Syk tSH2), green, and a receptor tail, purple, was detected by rhodamine quenching of BG505 dye on the reporter (see Materials and methods). Kd was determined by assessing mean fluorescence quenching for the last 20 timepoints collected ~45 min after ATP addition over a receptor titration from 0 to 500 nM. Each point represents the mean ± SD from three independent experiments. Kd ± SE was calculated by nonlinear fit assuming one site specific binding.

DOI: https://doi.org/10.7554/eLife.36688.007

The following figure supplement is available for figure 3:

**Figure supplement 1.** F-actin is enriched at the cell-target synapse.
DOI: https://doi.org/10.7554/eLife.36688.008

robust, suggesting that productive macrophage target interactions were frequently insufficient to trigger whole cell engulfment. To determine if whole cell eating could be enhanced by further opsonization of CD19, we opsonized Raji B cells with a mouse IgG2a anti-CD19 antibody. While addition of this antibody did not trigger additional whole cell internalization, blockade of the 'don't eat me' signal CD47 using the mouse IgG1 anti-human B6H12 clone resulted in a 2.5 fold increase of whole cell eating of opsonized Raji B cells (*Figure 4—figure supplement 3*). Both endogenous FcR recognition of the anti-CD47 antibody and blockade of CD47 signaling may contribute to this effect.

To develop a receptor to enhance whole cell eating, we hypothesized that combining signaling motifs in a tandem array might increase the frequency of whole cell engulfment by specifically recruiting effectors required for the engulfment of large targets. Previous work demonstrated that PI3K signaling is important for internalization of large targets (*Schlam et al., 2015*). To increase PI3K

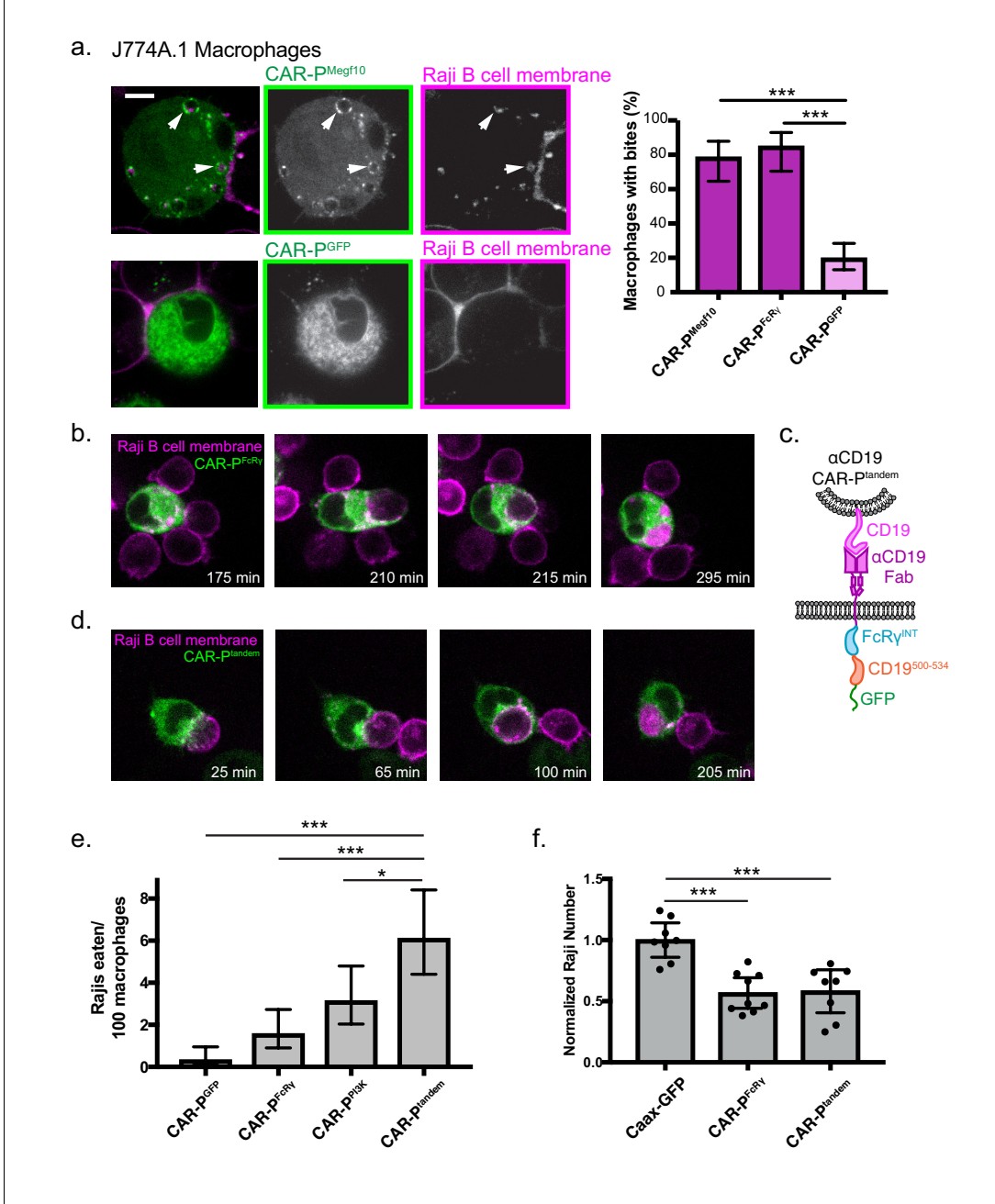

**Figure 4.** CAR-P promotes trogocytosis and whole cell eating. (**A**) J774A.1 macrophages expressing the αCD19 CAR-P^Megf10 (top panel, green in merge, left; greyscale, center) engulf pieces of CD19 +Raji B cells (labeled with mCherry-CAAX; magenta in merge, left; greyscale, right). The corresponding control αCD19 CAR-P^GFP-infected cells are shown below. Arrows point to pieces of ingested Raji B cell. The proportion of CAR-P expressing macrophages internalizing one or more bite within 90 min is quantified on the right. Bites are defined as a fully internalized mCherry-positive vesicle >1 μm in diameter; n = 46 CAR-P^Megf10 macrophages, n = 39 CAR-P^FcRγ macrophages and 102 CAR-P^GFP macrophages acquired during three separate experiments. (**B**) Time course of a J774A.1 macrophage expressing CAR-P^FcRγ (green) internalizing a whole Raji B cell labeled with mCherry-CAAX (magenta). These images correspond to frames from *Figure 4—video 2*. (**C**) Schematic shows the structure of CAR-P^tandem construct, combining the intracellular signaling domain from FcRγ and the p85 recruitment domain from CD19. (**D**) Time course of a J774A.1 macrophage expressing CAR-P^tandem (green) internalizing a whole Raji B cell labeled with mCherry-CAAX (magenta). These images correspond to frames from *Figure 4—video 3*. (**E**) Macrophages and Raji B cells were incubated together at a 1:2 macrophage:Raji ratio, and the number of whole Raji B cells eaten per 100 macrophages during 4–8 hr of imaging is graphed. Graph depicts pooled data from four independent experiments; n = 921 CAR-P^GFP, n = 762 CAR-P^FcRγ, n = 638 CAR-P^PI3K, n = 555 CAR-P^tandem cells. Sample sizes were selected for their ability to detect a 5% difference between samples with 95% confidence. (**F**) 10,000 macrophages and 20,000 Raji B cells were incubated together for 44 hr. The number of Rajis was then

*Figure 4 continued on next page*

*Figure 4 continued*

quantified by FACS. 2–3 technical replicates were acquired each day on three separate days. The number of Rajis in each replicate was normalized to the average number present in the GFP-CAAX macrophage wells on that day. * indicates p<0.01, *** indicates p<0.0001 by two-tailed Fisher Exact Test (a and e) or by Ordinary one way ANOVA with Dunnet's correction for multiple comparisons (f); error bars denote 95% confidence intervals.
DOI: https://doi.org/10.7554/eLife.36688.009

The following video and figure supplements are available for figure 4:

**Figure supplement 1.** CAR-P localizes with pTyr at synapse with Raji B cell.
DOI: https://doi.org/10.7554/eLife.36688.010

**Figure supplement 2.** NIH 3T3 cells internalize Raji B cell bites.
DOI: https://doi.org/10.7554/eLife.36688.011

**Figure supplement 3.** Opsonization by an anti-CD47 antibody enhances whole cell internalization through CAR-P Macrophages expressing CAR-P$^{FcR\gamma}$ and Raji B cells were incubated together at a 1:2 macrophage:Raji ratio (20,000 macrophages and 40,000 Rajis) without antibody addition (No ab) or in the presence of anti-CD19 or anti-CD47 antibodies as indicated.
DOI: https://doi.org/10.7554/eLife.36688.012

**Figure supplement 4.** CAR-P promotes internalization of cancer antigen.
DOI: https://doi.org/10.7554/eLife.36688.013

**Figure 4—video 1.** CAR-P$^{Megf10}$ macrophage engulfs bites of a Raji B cell.
DOI: https://doi.org/10.7554/eLife.36688.014

**Figure 4—video 2.** CAR-P$^{FcR\gamma}$ macrophage engulfs a Raji B cell.
DOI: https://doi.org/10.7554/eLife.36688.015

**Figure 4—video 3.** CAR-P$^{tandem}$ macrophage engulfs a Raji B cell.
DOI: https://doi.org/10.7554/eLife.36688.016

recruitment to the CAR-P, we fused the portion of the CD19 cytoplasmic domain (amino acids 500 to 534) that recruits the p85 subunit of PI3K to the CAR-P$^{FcR\gamma}$ creating a 'tandem' CAR (CAR-P$^{tandem}$, *Figure 4c*) (*Tuveson et al., 1993*; *Brooks et al., 2004*). A CAR-P containing the p85 recruitment motif alone (CAR-P$^{PI3K}$) was able to induce some whole cell engulfment, comparable to the CAR-P$^{FcR\gamma}$ (*Figure 4e*). Expression of CAR-P$^{tandem}$ tripled the ability of macrophages to ingest whole cells compared to CAR-P$^{GFP}$ (6 cancer cells eaten per 100 macrophages, *Figure 4d,e*, *Figure 4—video 3*). These data indicate that assembling an array of motifs designed to recruit distinct phagocytic effectors can increase CAR-P activity towards whole cells.

To determine if the combination of whole cell eating and trogocytosis was sufficient to drive a noticeable reduction in cancer cell number, we incubated CAR-P macrophages with Raji B cells for two days. After 44 hr of co-culture, we found that CAR-P macrophages significantly reduced the number of Raji cells (*Figure 4f*). Although the CAR-P$^{tandem}$ was much more efficient at whole cell eating, the CAR-P$^{FcR\gamma}$ performed nearly as well at eliminating Rajis. Importantly, our assay does not distinguish between whole cell engulfment or death following trogocytosis, so it is possible both CAR-P activities are contributing to Raji death rates. Overall, these data suggest that the CAR-P is a successful strategy for directing macrophages towards cancer targets, and can initiate whole cell eating and trogocytosis leading to cancer cell elimination.

In summary, we engineered phagocytes that recognize and ingest targets through specific antibody-mediated interactions. This strategy can be directed towards multiple extracellular ligands (CD19 and CD22) and can be used with several intracellular signaling domains that contain ITAM motifs (Megf10, FcR$\gamma$, and CD3$\zeta$). Previous work has suggested that spatial segregation between Src-family kinases and an inhibitory phosphatase, driven by receptor ligation, is sufficient to trigger signaling by the T cell receptor (*Davis and van der Merwe, 2006*; *James and Vale, 2012*) and FcR (*Freeman et al., 2016*). The CAR-Ps that we have developed may similarly convert receptor-ligand binding into receptor phosphorylation of ITAM domains through partitioning of kinases and phosphatases at the membrane-membrane interface.

Further development of CAR-Ps could be useful on several therapeutic fronts. Targeting of tumor cells by macrophages has been suggested to cause tumor cell killing (*Jaiswal et al., 2009*; *Majeti et al., 2009*; *Chao et al., 2010*; *Jadus et al., 1996*), either through directly engulfing cancer cells or by stimulating antigen presentation and a T cell-mediated response (*Liu et al., 2015*; *Tseng et al., 2013*). Inhibition of the CD47-SIRPA 'Don't eat me' signaling pathway has also been shown to result in engulfment of cancer cells (*Chen et al., 2017*; *Gardai et al., 2005*; *Jaiswal et al.,*

*2009*; *Majeti et al., 2009*; *Chao et al., 2010*). A recent study suggests that CD47 inhibition is most effective when combined with a positive signal to promote target engulfment, which raises the possibility of combining CAR-P expression with CD47 or SIRPA inhibition for an additive effect (*Alvey et al., 2017*).

Although we were able to increase whole cell engulfment by recruiting the activating subunit of PI3K to the phagocytic synapse, the engulfment of larger 20 micron beads was more frequent than the engulfment of whole cells. We hypothesize that this is due to differing physical properties of the engulfment target. Specifically, increased target stiffness has been shown to promote engulfment, suggesting that manipulating the physical properties of the engulfment target could also be a potential strategy for increasing CAR-P efficiency (*Beningo and Wang, 2002*; *Cross et al., 2007*).

While the CAR-P can engulf whole, viable cancer cells, the ingestion of a piece of the target cell is more common. Trogocytosis, or nibbling of living cells, has also been described in immune cells (*Harshyne et al., 2003*; *Harshyne et al., 2001*; *Kao et al., 2006*; *Joly and Hudrisier, 2003*; *Batista et al., 2001*) and brain-eating amoebae (*Ralston et al., 2014*). In vivo studies also have shown that endogenous dendritic cell populations ingest bites of live tumor cells, contributing to presentation of cancer neo-antigen (*Harshyne et al., 2001*; *Harshyne et al., 2003*). Although we were able to use the CAR-P to induce trogocytosis in dendritic cells, we were not able to detect robust cross presentation of the model antigen ovalbumin (*Figure 4—figure supplement 4*). Thus, although using CAR-Ps to enhance cross presentation of cancer antigen is an intriguing future avenue, such a strategy would likely require more optimization of the dendritic cell subset employed or the CAR-P receptor itself.

Overall, our study demonstrates that the CAR approach is transferrable to biological processes beyond T cell activation and that the expression of an engineered receptor in phagocytic cells is sufficient to promote specific engulfment and elimination of cancer cells.

# Materials and methods

**Key resources table**

| Reagent type (species) or resource | Designation | Source or reference | Identifiers | Additional information |
|---|---|---|---|---|
| Cell line (*Mus musculus*) | J774A.1 Macrophages | UCSF Cell Culture Facility | | |
| Cell line (*Homo sapiens*) | Raji B Cells | Other | | Obtained from M. McManus, UCSF |
| Cell line (*Mus musculus*) | 3t3 Fibroblasts | UCSF Cell Culture Facility | | |
| Cell line (*Mus musculus*) | C57BL/6J | PMID: 21356739 | | Bone Marrow Derived Macrophages (BMDM) |
| Cell line (*Mus musculus*) | C57BL/6J | PMID: 7489412 | | Bone Marrow derived Dendritic Cells (BMDC) |
| Cell line (*Homo sapiens*) | HEK293T cells | UCSF Cell Culture Facility | | Lentivirus production |
| Genetic Reagent (*Mus musculus*) | OTI | PMID: 8287475 | | E. Roberts/M. Krummel Lab UCSF |
| Recombinant DNA reagent | CD19-mMegf10 CAR | this paper | | Signal peptide: aa 1–21 CD8 (Uniprot Q96QR6_HUMAN) Extracellular antibody sequence: V-L chain: aa 23–130 anti-CD19 CAR (Genbank AMZ04819) – GS linker: ggtggcggtggctcgggcggtggtgggtcgggt ggcggcggatct – V-H chain: aa 148–267 anti-CD19 CAR (Genbank AMZ04819) Stalk/Transmembrane: aa 138–206 CD8 (Uniprot Q96QR6_HUMAN) Cytosolic sequence: aa 879–1147 Mouse Megf10 (Uniprot Q6DIB5 (MEG10_MOUSE)) Fluorophore: mGFP |

*Continued on next page*

*Continued*

| Reagent type (species) or resource | Designation | Source or reference | Identifiers | Additional information |
|---|---|---|---|---|
| Recombinant DNA reagent | CD19-FcGamma CAR | this paper | | Signal peptide: aa 1–21 CD8 (Uniprot Q96QR6_HUMAN) Extracellular antibody sequence: V-L chain: aa 23–130 anti-CD19 CAR (Genbank AMZ04819) – GS linker: ggtggcggtggctcgggcggtggtgggtcgg gtggcggcggatct – V-H chain: aa 148–267 anti-CD19 CAR (Genbank AMZ04819) Stalk/Transmembrane: aa 138–206 CD8 (Uniprot Q96QR6_HUMAN) Cytosolic sequence: aa 19–86 Mouse Fc ERG precursor (Uniprot P20491 (FCERG_MOUSE)) Fluorophore: mGFP |
| Recombinant DNA reagent | CD19-empty CAR | this paper | | Signal peptide: aa 1–21 CD8 (Uniprot Q96QR6_HUMAN) Extracellular antibody sequence: V-L chain: aa 23–130 anti-CD19 CAR (Genbank AMZ04819) – GS linker: ggtggcggtggctcgggcggtggtgggtcggg tggcggcggatct – V-H chain: aa 148–267 anti-CD19 CAR (Genbank AMZ04819) Stalk/Transmembrane: aa 138–206 CD8 (Uniprot Q96QR6_HUMAN) Cytosolic sequence: basic linker NHRNRRR (nucleotide AACCACAGG AACCGAAGACGT) Fluorophore: mGFP |
| Recombinant DNA reagent | CD22-Megf10 CAR | this paper | | Signal peptide: aa 1–21 CSF2R (Uniprot P15509 (CSF2R_HUMAN)) Extracellular antibody sequence: aa 22–258 of translated JP 2016502512-A/1: M971 Chimeric Antigen (Genbank HZ530416.1) Stalk/Transmembrane: aa 138–206 CD8 (Uniprot Q96QR6_ HUMAN) Cytosolic sequence: aa 879–1147 Mouse Megf10 (Uniprot Q6DIB5 (MEG10_ MOUSE)) Fluorophore: mGFP |
| Recombinant DNA reagent | CD22-empty CAR | this paper | | Signal peptide: aa 1–21 CSF2R (Uniprot P15509 (CSF2R_HUMAN)) Extracellular antibody sequence: aa 22–258 of translated JP 2016502512-A/1: M971 Chimeric Antigen (Genbank HZ530416.1) Stalk/Transmembrane: aa 138–206 CD8 (Uniprot Q96QR6_HUMAN) Cytosolic sequence: basic linker NHRNRRR (nucleotide AACCACAGGAACCGAAGACGT) Fluorophore: mGFP |
| Recombinant DNA reagent | CD19-MerTK CAR | this paper | | Signal peptide: aa 1–21 CD8 (Uniprot Q96QR6 _HUMAN) Extracellular antibody sequence: V-L chain: aa 23–130 anti-CD19 CAR (Genbank AMZ04819) – GS linker: ggtg gcggtggctcgggcggtggtgggtcgggtggcggcggatct – V-H chain: aa 148–267 anti-CD19 CAR (Genbank AMZ04819) Stalk/Transmembrane: aa 138–206 CD8 (Uniprot Q96QR6_HUMAN) Cytosolic sequence: aa 519–994 Mouse MerTK (Uniprot Q60805 (MERTK_MOUSE)) Fluorophore: mGFP |
| Recombinant DNA reagent | CD19-Bai1 CAR | this paper | | Signal peptide: aa 1–21 CD8 (Uniprot Q96QR6_HUMAN)Extracellular antibody sequence: V-L chain:aa 23–130 anti-CD19 CAR (Genbank AMZ04819)– GS linker: ggtggcggtggctcgggcggtggtgggtcgggtggcgg cggatct – V-H chain: aa 148–267 anti-CD19 CAR (GenbankAMZ04819) Stalk/Transmembrane: aa 138–206 CD8 (UniprotQ96QR6_HUMAN) Cytosolic sequence: aa1188–1582 Mouse Bai1 (Uniprot Q3UHD1 (BAI1_MOUSE)) Fluorophore: mGFP |

*Continued on next page*

Continued

| Reagent type (species) or resource | Designation | Source or reference | Identifiers | Additional information |
|---|---|---|---|---|
| Recombinant DNA reagent | CD19-CD3zeta CAR | this paper | | Signal peptide: aa 1–21 CD8 (Uniprot Q96QR6_HUMAN) Extracellular antibody sequence: V-L chain: aa 23–130 anti-CD19 CAR (Genbank AMZ04819) – GS linker: ggtggcggtggctcg ggcggtggtgggtcgggtggcggcggatct – V-H chain: aa 148–267 anti-CD19 CAR (Genbank AMZ04819) Stalk/Transmembrane: aa 138–206 CD8 (Uniprot Q96QR6_HUMAN) Cytosolic sequence: aa 52–164, Human TCR CD3 zeta chain (Uniprot P20963) Fluorophore: sfGFP |
| Recombinant DNA reagent | CD19-PI3K CAR | this paper | | Signal peptide: aa 1–21 CD8 (Uniprot Q96QR6_HUMAN) Extracellular antibody sequence: V-L chain: aa 23–130 anti-CD19 CAR (Genbank AMZ04819) – GS linker: ggtggcggtggct cgggcggtggtgggtcgggtggcggcggatct – V-H chain: aa 148–267 anti-CD19 CAR (Genbank AMZ04819) Stalk/Transmembrane: aa 138–206 CD8 (Uniprot Q96QR6_HUMAN) Cytosolic sequence: aa 500–534 Mouse CD19 (Uniprot CD19_MOUSE) Fluorophore: mCherry |
| Recombinant DNA reagent | CD19 tandem CAR | this paper | | Signal peptide: aa 1–21 CD8 (Uniprot Q96QR6_HUMAN) Extracellular antibody sequence: V-L chain: aa 23–130 anti-CD19 CAR (Genbank AMZ04819) – GS linker: ggtggcggtggctc gggcggtggtgggtcgggtggcggcggatct – V-H chain: aa 148–267 anti-CD19 CAR (Genbank AMZ04819) Stalk/Transmembrane: aa 138–206 CD8 (Uniprot Q96QR6_HUMAN) Cytosolic sequence: aa 500–534 Mouse CD19 (Uniprot CD19_MOUSE) fused to aa 19–86 Mouse Fc ERG precursor (FCERG_MOUSE) Fluorophore: mGFP |
| Recombinant DNA reagent | GFP-CaaX | this paper | | eGFP fused to a c terminal CaaX targeting sequence: aaaatgtccaaggatggta agaaaaagaagaagaagtcaaaaaccaagtgtgttatcatg |
| Recombinant DNA reagent | mCherry-CaaX | this paper | | mCherry fused to a c terminal CaaX targeting sequence: aaaatgtccaaggatggt aagaaaaagaagaagaagtcaaaaaccaagtgtgttatcatg |
| Recombinant DNA reagent | OVA/p2a/mCherry-CaaX | this paper | | Cytoplasmic Ovalbumin (UNIPROT: SERPINB14)/p2 a site: GGAAGCGGAGCTACTAA CTTCAGCCTGCTGAAGCAGGCTGGAGA CGTGGAGGAGAACCCTGGACCT/followed by mCherry fused to a c terminal CaaX targeting sequence: aaaatgtccaaggatggtaagaaaaagaag aagaagtcaaaaaccaagtgtgttatcatg |
| Peptide, recombinant protein | His10-CD3 zeta | *Hui and Vale (2014)* PMID: 24463463 | | aa 52–164, Human TCR CD3 zeta chain (Uniprot CD3Z_HUMAN) fused to Hisx10 tag |
| Peptide, recombinant protein | His10-FcRγ | this paper | | aa 45–85, Human FcRγ (Uniprot FCERG_HUMAN) fused to Hisx10 tag |
| Peptide, recombinant protein | SNAP-Syk tSH2 | this paper | | aa 1–262, Mouse Syk (Uniprot KSYK_MOUSE) with N-term SNAP tag |
| Peptide, recombinant protein | His10-Lck Y505F | *Hui and Vale (2014)* PMID: 24463463 | | full length Human Lck with inhibitory Tyr 505 mutated to Phe (Uniprot LCK_HUMAN) fused to Hisx10 tag |
| Antibody | anti phospho-Tyrosine | Santa Cruz | PY20 | 1:100 IF primary |
| Antibody | anti mouse IgG coupled to Alexa Fluor 647 | Thermo/Lifetech | A21236 | 1:200 IF secondary |
| Antibody | anti mouse CD11c coupled to APC | BioLegend | 117313 | FACS |

Continued

| Reagent type (species) or resource | Designation | Source or reference | Identifiers | Additional information |
|---|---|---|---|---|
| Antibody | anti mouse F4/80 coupled to APC/Cy7 | BioLegend | 123117 | FACS |
| Other | DMEM | Gibco | 11965–092 | |
| Other | Pen-Strep-Glutamine | Corning | 30–009 Cl | |
| Other | Fetal Bovine Serum (FBS) | Atlanta Biologicals | S1150H | |
| Other | RPMI | Gibco | 11875–093 | |
| Other | HEPES | Gibco | 1530080 | |
| Other | 2-Mercaptoethanol | Sigma | M6250-100mL | |
| Commercial assay or kit | MycoAlert Mycoplasma Testing Kit | Lonza | LT07-318 | |
| Recombinant DNA reagent | pMD2.G lentiviral plasmid | other | Addgene 12259 | D. Stainer, Max Planck; VSV-G envelope |
| Recombinant DNA reagent | pCMV-dR8.91 | other | Current Addgene 8455 | |
| Recombinant DNA reagent | pHRSIN-CSGW | other | | As cited *James and Vale (2012)*, PMID: 22763440 |
| Other | Lipofectamine LTX | Invitrogen | 15338–100 | Lentivirus production |
| Other | Lipofectamine | Invitrogen | 18324–012 | Added to spin infections to improve transduction |
| Other | Hamilton Gastight Syringes | Hamilton | 8 1100 | |
| Other | POPC | Avanti | 850457 | |
| Other | Ni2+-DGS-NTA | Avanti | 790404 | |
| Other | PEG5000-PE | Avanti | 880230 | |
| Other | atto390 DOPE | ATTO-TEC GmbH | AD 390–161 | |
| Other | PBS (Tissue Culture Grade) | Gibco | 20012050 | |
| Other | Bioruptor Pico | Diagenode | | Used for producing SUVs |
| Other | 5 um silica microspheres | Bangs | SS05N | |
| Peptide, recombinant protein | CD19-His8 | Sino Biological | 11880H08H50 | |
| Peptide, recombinant protein | CD22-His8 | Sino Biological | 11958H08H50 | |
| Other | 2.5 um silica microspheres (size titration) | Corpuscular | C-SIO-2.5 | |
| Other | 5 um silica microspheres (size titration) | Corpuscular | C-SIO-5 | |
| Other | 10 um silica microspheres (size titration) | Corpuscular | C-SIO-10 | |
| Other | 15 um silica microspheres (size titration) | Corpuscular | C-SIO-15 | |
| Other | 20 um silica microspheres (size titration) | Corpuscular | C-SIO-20 | |
| Other | Low retention tubes for microsphere cleaning | Eppendorf | 22431081 | |
| Other | MatriPlate | Brooks | MGB096-1-2-LG-L | |
| Peptide, recombinant protein | M-CSF | Peprotech | 315–02 | |
| Other | IMDM | Thermo | 12440079 | |

Continued

| Reagent type (species) or resource | Designation | Source or reference | Identifiers | Additional information |
|---|---|---|---|---|
| Other | Retronectin | Clontech | T100A | |
| Commercial assay or kit | CD8 + T cell purification kit | Stemcell | 19853 | |
| Other | eFluor670 proliferation dye | Thermo | 65-0840-85 | |
| Chemical compound, drug | phRSIN-CSGW | Sigma | L4516 | |
| Other | Fluorobrite DMEM | Gibco | A1896701 | |
| Other | DMEM minus phenol red | Gibco | A14430-01 | |
| Other | Rhodamine PE | Avanti | 810150C | |
| Other | DOPS | Avanti | 840035C | |
| Other | SNAP-Cell 505-Star | NEB | S9103S | |
| Other | PD MiniTrap G-25 column | GE Healthcare | 28-9225-29 AB | |
| Other | 6.4% Paraformaldehyde solution | Electron Microscopy Sciences | 50980495 | |
| Chemical compound, drug | AlexaFluor 647 Phalloidin | Thermo/Molecular Probes | A22284 | |
| Software, algorithm | ImageJ | NIH | | |
| Software, algorithm | Illustrator | Adobe | CC, CS6 | |
| Software, algorithm | Photoshop | Adobe | CC, CS6 | |
| Software, algorithm | Fiji | https://fiji.sc/ | | |
| Software, algorithm | Prism | GraphPad | 7 | |
| Antibody | anti human CD19 (mouse antibody) | OriGene | TA506240 Clone OTI2F6 | IgG2a mouse monoclonal antibody |
| Antibody | anti human CD47 (mouse antibody) | BD | 556044 Clone B6H12 | IgG1 mouse monoclonal antibody |
| Antibody | anti Ovalbumin (rabbit antibody) | Pierce | PA1-196 | IgG rabbit polyclonal antibody |

## Constructs and antibodies

Detailed information for all constructs can be found in the Key resources table. This file includes the following information for all receptors developed in this study: signal peptide, extracellular antibody fragment, stalk/transmembrane domain, and cytosolic tail including appropriate accession numbers. Antibodies used in this study are described in Supplemental Excel File 1, 'Antibodies' tab.

## Cell culture

J774A.1 macrophages and NIH 3T3 fibroblasts were obtained from the UCSF cell culture facility and cultured in DMEM (Gibco, Catalog #11965–092) supplemented with 1 x Pen-Strep-Glutamine (Corning, Catalog #30–009 Cl) and 10% fetal bovine serum (FBS) (Atlanta Biologicals, Catalog #S11150H). Raji B cells were obtained from J. Blau (McManus lab, UCSF) and cultured in RPMI (Gibco, Catalog #11875–093) supplemented with 1 x Pen-Strep-Glutamine (Corning, Catalog #30–009 Cl), 10% FBS (Atlanta Biologicals, Catalog #S11150H), 10 mM HEPES (Gibco, Catalog #1530080), and 5 µM 2-Mercaptoethanol (Sigma, Catalog #M6250-100mL). All cell lines used in this study were tested for Mycoplasma at least once per month using the Lonza MycoAlert Detection Kit (Lonza, Catalog# LT07-318) and control set (Lonza, Catalog #LT07-518).

## Lentivirus production and infection

Lentiviral infection was used to stably express CAR-P constructs in all cell types. Lentivirus was produced by HEK293T cells transfected with pMD2.G (a gift from Didier Tronon, Addgene plasmid # 12259 containing the VSV-G envelope protein), pCMV-dR8.91 (since replaced by second generation compatible pCMV-dR8.2, Addgene plasmid #8455), and a lentiviral backbone vector containing the construct of interest (derived from pHRSIN-CSGW) using lipofectamine LTX (Invitrogen, Catalog # 15338–100). The media on the HEK293T cells was replaced with fresh media 8–16 hr post transfection to remove transfection reagent. At 50–72 hr post-transfection, the lentiviral media was filtered with a 0.45 µm filter and concentrated by centrifugation at 8000 x g for 4 hr or overnight. The concentrated supernatant was applied directly to ~$0.5{\times}10^6$ NIH 3T3 cells in 2 ml of fresh media. For J77A4.1 macrophages and Raji B cells, the concentrated supernatant was mixed with 2 mls of media and 2 µg lipofectamine (Invitrogen, Catalog # 18324–012) and added to the cells. The cells were spun at 2200 x g for 45 min at 37°C. Cells were analyzed a minimum of 72 hr later.

## Preparation of CD19 and CD22 5 µm silica beads

Chloroform-suspended lipids were mixed in the following molar ratio using clean glasstight Hamilton syringes (Hamilton, Catalog #8 1100): 97% POPC (Avanti, Catalog # 850457), 2% Ni2+-DGS-NTA (Avanti, Catalog # 790404), 0.5% PEG5000-PE (Avanti, Catalog # 880230, and 0.5% atto390-DOPE (ATTO-TEC GmbH, Catalog # AD 390–161). Lipid mixes were dried under argon and desiccated overnight under foil. Dried lipids were resuspended in 1 ml tissue-culture grade PBS, pH7.2 (Gibco, Catalog # 20012050), and stored under argon gas. Small unilamellar vesicles were formed by five freeze-thaw cycles followed by 2 × 5 min of bath sonication (Bioruptor Pico, Diagenode), and cleared by ultracentrifugation (TLA120.1 rotor, 35,000 rpm / 53,227 x g, 35 min, 4°C) or by 33 freeze thaw cycles. Lipid mixes were used immediately for form bilayers or shock frozen in liquid nitrogen and stored under argon at −80°C. To form bilayers on silica beads, 6 × $10^8$ 5 µm silica microspheres (10% solids, Bangs Labs, Catalog # SS05N) were washed 2x in water, and 2x in PBS by sequential suspension in water and spinning at 800 rcf, followed by decanting. Cleaned beads were resuspended in 150 µl tissue-culture grade PBS, pH7.2 (Gibco, Catalog # 20012050) and briefly vortexed. 30 µl cleared SUVs prepared as above as a 10 mM stock were added to bead suspension for a 2 mM final SUV concentration. Beads were vortexed for 10 s, covered in foil, and rotated for 30 min at room temperature to form bilayers. Bilayer-coated beads were washed 3x in PBS by sequential centrifugation at 800 rcf and decanting. Beads were resuspended in PBS + 0.1% w/v BSA for blocking for 15 min rotating at room temperature under foil. 10 nM final concentration of $CD19_{his8}$ (Sino Biological, Catalog # 11880H08H50) or $CD22_{his8}$ (Sino Biological, Catalog # 11958H08H50) protein were added to blocked beads and proteins were allowed to bind during a 45 min incubation rotating under foil at room temperature. Beads were washed 3x in PBS + 0.1% w/v BSA by sequential centrifugation at 300 rcf and decanting. Beads were resuspended in 120 µl PBS + 0.1% w/v BSA.

## Preparation of CD19 silica beads over a range of diameters

Prior building bilayers on Silica beads ranging from 2.5 µm-20 µm in diameter (Microspheres-Nanospheres, Catalog# C-SIO-2.5, 5, 10, 15, 20), beads were RCA cleaned as follows: beads were pelleted at 2000 x g in low retention tubes (Eppendorf, Catalog #022431081) and resuspended in acetone. Resuspended beads were sonicated for 60 min in a bath sonicator. Rinse and sonication were repeated in ethanol. Finally, rinse and sonication were repeated in water. Beads were then washed 2x in water to remove all traces of ethanol and left in a small volume after decanting. All further steps were performed in a 70–80°C water bath prepared in a fume hood. Proper Personal Protective Equipment (PPE) was worn throughout the RCA cleaning protocol. Washed beads were added to 3 ml of hot 1.5 M KOH in a clean glass vial suspended in the water bath described above. 1 ml 30% $H_2O_2$ to bead solution and allowed to react for 10 min. Washed beads were cooled on ice, pelleted at 2000xg and rinsed 5x in ultrapure water. Used cleaning solution was saved for disposal by Environmental Health and Safety (EH and S). Cleaned beads were resuspended in 240 µl tissue-culture grade PBS, pH7.2 (Gibco, Catalog # 20012050) and briefly vortexed. The lipid mix used in this assay differed slightly from above. Here a mix of 93.5% POPC (Avanti, Catalog # 850457), 5% $Ni^{2+}$-DGS-NTA (Avanti, Catalog # 790404), 1% PEG5000-PE (Avanti, Catalog # 880230, and 0.5% atto390-DOPE (ATTO-TEC GmbH, Catalog # AD 390–161. Bilayers were built and proteins coupled

as described above. The concentration of CD19 was scaled appropriately to account for the increased surface area of the larger beads.

## Bead engulfment assay

12 to 16 hr prior to imaging, $2.5 \times 10^4$ J774A.1 macrophages expressing the appropriate CAR-P or control construct were plated in a 96-well glass bottom MatriPlate (Brooks, Catalog # MGB096-1-2-LG-L). To assess engulfment, $0.5 \times 10^6$ CD19 or CD22-ligated beads were added to each well. Engulfment was allowed to proceed for 45 min at 37°C incubator with $CO_2$. Cells were then imaged as described below.

## Bites assay – J774A.1 macrophages, dendritic cells and NIH 3T3 fibroblasts

On the day of imaging $0.5 \times 10^6$ NIH 3T3 fibroblasts, dendritic cells or macrophages and 1.5 million Raji B cells were combined in a 1.5 ml eppendorf tube and pelleted by centrifugation (800 rpm / 68 x g) for 5 min at room temperature. Culture media was decanted to ~100 µl volume and cells were gently resuspended, and allowed to interact in the small volume for 60 min in a 37°C incubator with $CO_2$. After incubation cells and beads were diluted to a final volume of 1000 µl and 300 µl of this co-culture plated for imaging in a 96-well glass bottom MatriPlate (Brooks, Catalog # MGB096-1-2-LG-L), and imaged as described below.

## Eating assay read by FACS – J774A.1 macrophages and Raji B cells

20,000 J774A.1 macrophages were plated into 96-well glass bottom MatriPlate (Brooks, Catalog # MGB096-1-2-LG-L) in a final volume of 300 µl complete DMEM (Gibco, Catalog #11965–092) supplemented with 1 x Pen-Strep-Glutamine (Corning, Catalog #30–009 Cl) and 10% fetal bovine serum (FBS) (Atlanta Biologicals, Catalog #S11150H). 52 hr prior to reading the assay macrophages were stimulated with 500 ng/ml LPS (Sigma, Catalog # L4516). 44 hr prior to imaging LPS was removed by three sequential gentle washes. After LPS removal 10,000 Rajis expressing mCherry-CAAX were added to the well containing stimulated macrophages. The co-culture was incubated for 44 hr in a 37°C tissue culture incubator with 5% $CO_2$. After 44 hr, the remaining number of Raji B cells remaining was analyzed by FACS as follows: 10,000 counting beads were added to the well immediately prior to reading and the cell-counting bead mixture was harvested by pipetting up and down 8x with a p200 pipet. The assay was read on an LSRII (BD Biosciences) and Rajis were identified by the presence of mCherry fluorescence.

## Primary cell transduction and differentiation

Bone marrow derived macrophages (BMDMs) were produced as previously described (*Weischenfeldt and Porse, 2008*), except that L-929 conditioned media was replaced with purified 25 ng/ml M-CSF (Peprotech, Catalog # 315–02). The BMDMs were lentivirally infected with concentrated lentivirus after 5 days of differentiation. Differentiation was confirmed by F4/80 staining on day seven and found to be >95% efficient for each replicate. Phagocytosis was measured on day nine in imaging media lacking M-CSF.

To produce CAR-P expressing dendritic cells, bone marrow-derived hematopoietic stem cells were lentivirally infected immediately after harvest by spinning with concentrated lentivirus in GMCSF-containing media (IMDM supplemented with 10% FBS and PSG) on retronectin (Clontech, Catalog # T100A)-coated plates at 2200 x g for 45 min at 37°C. Dendritic cells were produced as previously described (*Mayordomo et al., 1995*) by culturing bone marrow cells for 8–11 days with GMCSF. IL-4 was added 2–3 days before use. Efficient differentiation into CD11c + dendritic cells was verified by FACS, revealing ≥95% APC-CD11c + cells (Biolegend, Catalog #N418).

## Antigen cross-presentation assay

The ability of CAR-P to stimulate OTI T cell proliferation was tested using the co-culture assay shown as a schematic in *Figure 4—figure supplement 4* and described previously (*Roberts et al., 2016*). 10,000 CAR-P transduced CD11c + dendritic cells transduced and differentiated as above were plated in U bottom 96 well dishes (Falcon, Catalog #353077) and stimulated with 1 ug/ml LPS. 12 hr after LPS stimulation, 40,000 Raji B cells expressing soluble cytosolic ovalbumin (Raji B-OVA) were

added to the culture. 24 hr after Raji B-OVA cell addition, 50,000 OTI CD8 + T cells isolated from lymph nodes of OTI TCR transgenic mice using a CD8 +T cell purification kit (Stemcell, Catalog #19853) and labeled with e670 proliferation dye (Thermo, Catalog #65-0840-85) were added. 72 hr after OTI addition the percent of OTI cells divided was measured by eFluor670 signal using flow cytometry.

## Confocal imaging

All imaging in this study was performed using a spinning disk confocal microscope with environmental control (Nikon Ti-Eclipse inverted microscope with a Yokogawa spinning disk unit). For bead internalization assays, images were acquired using a 40 × 0.95 N/A air objective and unbiased live image acquisition was performed using the High Content Screening (HCS) Site Generator plugin in µManager[3]. Other images were acquired using either a 100 × 1.49 N/A oil immersion objective. All images were acquired using an Andor iXon EM-CCD camera. The open source µManager software package was used to control the microscope and acquire all images[3].

## Quantification of whole cell internalization

20,000 J774A.1 macrophages were plated into 96-well glass bottom MatriPlate (Brooks, Catalog # MGB096-1-2-LG-L). Four hours prior to imaging, the macrophages were stimulated with 500 ng/ml LPS (Sigma, Catalog # L4516). Immediately prior to imaging the LPS-containing media was replaced with Fluobrite DMEM (ThermoFisher Scientific, Catalog # A1896701) containing 10% FBS. 40,000 Raji cells were added to the macrophages and the co-culture was imaged at 5 min intervals for 12 hr. For the antibody experiments, macrophages were washed into DMEM minus phenol red (A14430-01) containing 10% FBS just prior to addition of 40,000 Raji cells. Where indicated antibody was added to a final concentration of 20 µg/ml immediately after Raji cell addition and prior to imaging to limit antibody internalization. Because cells moved in and out of the field of view, we selected the cells present after 8 hr of imaging and quantified their B cell eating if they could be followed for four hours or more. Time-lapse analysis was essential to ensure that the B cell appeared viable prior to engulfment by the macrophage. Engulfment of B cells with an apoptotic morphology was not counted as a whole cell eating event.

## Quantification of bites internalization

During live cell image acquisition GFP-positive J774A.1 macrophages or NIH 3T3 cells were selected by the presence of GFP signal. A full z-stack comprising the entire cell was captured using 1 µm steps. All z sections were then manually inspected for internalized Raji B cell material. Cells containing one or more bites of fully internalized Raji B cell material >1 µm in diameter were scored as positive.

## Liposome FRET assay

Experiments were carried out as previously described[2]. Briefly, proteins were purified using a bacterial expression system. All protein components (1 mg/ml BSA, 100 nM tSH2-Syk SNAP-505, 0 to 500 nM His10-CD3ζ or His10-FcRγ intracellular chain, and 7.2 nM His10-Lck Y505F) were mixed into kinase buffer (50 mM HEPES-NaOH pH 6.8, 150 mM NaCl, 10 mM MgCl$_2$, and 1 mM TCEP). Liposomes prepared at the following molar ratios: 74.5% POPC (Avanti, Catalog # 850457C), 10% DOGS-NTA (Nickel) (Avanti, Catalog # 790404C, 0.5% Rhodamine PE (Avanti, Catalog # 810150C), and 15% DOPS (Avanti, Catalog # 840035C) were added and the mixture was incubated for 40–60 min at room temperature, during which the SNAP-505 fluorescence was monitored at 8 s intervals with 504 nm excitation and 540 nm emission. 1 mM ATP was then injected to trigger Lck mediated phosphorylation of CD3ζ or FcRγ. Injection was followed by 5 s of automatic shaking of the plate, and the fluorescence was further monitored at 8 s intervals for at least 1 hr. Data were normalized by setting the average fluorescence value of the last 10 data points before ATP addition as 100% and background fluorescence as 0%. The final extent of fluorescence quenching (% fluorescence change) at each concentration of receptor was determined using the average of the last 20 data points after ensuring fully equilibrated binding. Nine reactions containing increasing concentrations of CD3ζ and nine reactions containing increasing concentrations of FcRγ were run in parallel. The final % fluorescence change was plotted against FcRγ or CD3ζ concentration. The apparent

dissociation constants (Kd) of tSH2-Syk to FcRγ and CD3ζ were calculated by fitting the data with Graphpad Prism 6.0, using the 'one site specific binding' model.

## Protein expression, purification, and labeling

The intracellular portion of the FcR γ-chain (aa 45–85, Human FcRγ, Uniprot FCERG_HUMAN) was cloned into a modified pET28a vector containing a His10 upstream to the multiple cloning site using BamHI and EcoRI. The intracellular portion of CD3ζ (aa 52–164, Human CD3ζ, Uniprot CD3Z_HUMAN) was also cloned into the His10 modified pET28a vector. A Lys-Cys-Lys-Lys sequence, originally present for fluorescent labeling, is also present between His10 and CD3ζ in this construct. SNAP-tSH2Syk (aa 1–262) was cloned into a pGEX6 vector using BamHI and EcoRI. His10-CD3ζ, $His_{10}$-FcR γ-chain, and GST-SNAP-tSH2Syk were bacterially expressed in BL21 (DE3) RIPL strain of *Escherichia coli* as described previously[2]. $His_{10}$-Lck Y505F was expressed in SF9 cells using the Bac-to-Bac baculovirus system as described previously[2]. All cells were lysed in an Avestin Emulsiflex system. $His_{10}$ proteins were purified by using Ni-NTA agarose (Qiagen, Catalog # 30230) and GST-SNAP-tSH2Syk was purified by using glutathione-Sepharose beads (GE Healthcare, Catalog # 17075601) as described previously[2]. Soluble SNAP-tSH2 Syk was generated by cleaving the GST moiety via the PreScission Protease at 4°C overnight. All proteins were subjected to gel-filtration chromatography using a Superdex 200 10/300 GL column (GE Healthcare, Catalog # 17517501) in HEPES-buffered saline (HBS) containing 50 mM HEPES-NaOH (pH 6.8 for $His_{10}$-CD3ζ, $His_{10}$-FcR γ-chain, and GST-SNAP-tSH2Syk and pH 7.4 for $His_{10}$-Lck Y505F), 150 mM NaCl, 5% glycerol, and 1 mM TCEP. The monomer fractions were pooled, frozen in liquid nitrogen and stored at −80°C. All gel-filtered proteins were quantified by SDS-PAGE and Coomassie staining, using BSA as a standard. To prepare fluorescently labeled tSH2 Syk, 10 uM SNAP-tSH2 Syk was incubated at a 1:2 ratio with SNAP-Cell 505-Star (NEB, Catalog # S9103S) overnight at 4°C and run over a PD MiniTrap G-25 (GE Healthcare, Catalog # 28-9225-29 AB) column to eliminate excess dye.

## Phosphotyrosine and phalloidin staining

To fix and stain preparations described above in bead and bites assays for quantifying enrichment of phosphotyrosine staining, half the media (~150 μl) was gently removed from the imaging well and replaced with 150 μl 6.4% paraformaldehyde solution (prepared from 32% stock, Electron Microscopy Sciences, Catalog # 50980495) in tissue culture grade PBS, pH7.2 (Gibco, Catalog # 20012050). Cells were fixed for 15 min in a 37°C incubator with $CO_2$. After fixation cells were washed 2x with PBS and permeabilized/blocked for 60 min at room temperature in freshly prepared, filter sterilized PBS + 5% FBS+0.1% w/v saponin (PFS solution). After permeabilization, cells were washed 2 × 3 min with PFS solution. Following block, cells were incubated with 1:100 dilution of mouse anti-phosphotyrosine (pTyr) antibody to stain pan-pTyr (Santa Cruz, Catalog # PY20) diluted in PFS solution in the dark for 60 min at room temperature then washed 3 × 5 min in PFS solution. Washed cells were incubated with a 1:500 dilution of goat anti-mouse Alexa Fluor 647 antibody (Thermo/Molecular Probes, Catalog # A21236) in PFS solution in the dark for 60 min at room temperature. Wells were then washed 3 × 5 min in PFS solution. Cells were covered in 200 μl PBS. If not imaged immediately samples were wrapped in parafilm and foil and stored at 4°C prior to microscopy. Phosphotyrosine enrichment at the synapse was calculated by dividing the mean Alexa Fluor 647 signal of a 5 pixel linescan at the synapse with bead or cell by a 5 pixel linescan on the cortex. For phalloidin staining, cells were fixed with 4% PFA for 15 min at room temperature, blocked and permeabilized with 5% BSA in TBS with 0.5% triton X overnight, and incubated with AlexaFluor 647 Phalloidin (Thermo/Molecular Probes, Catalog # A22284) for 20 min. Cells were then washed with PBS, imaged and quantified using the method described above. Each data point represents a single cell, and the graphs reflect pooled results from three biological replicates.

## Ovalbumin antibody staining

To fix and stain preparations described above for ovalbumin staining, half the media (~150 μl) was gently removed from the imaging well and replaced with 150 μl 8% paraformaldehyde solution (prepared from 32% stock, Electron Microscopy Sciences, Catalog # 50980495) in tissue culture grade PBS, pH7.2 (Gibco, Catalog # 20012050). Cells were fixed for 10 min in a 37°C incubator with $CO_2$. After fixation cells were washed 2x with PBS and permeabilized/blocked for 60 min at room

temperature in freshly prepared, filter sterilized PBS + 0.1% w/v casein +0.1% w/v saponin (PCS solution). After permeabilization, cells were washed $1 \times 3$ min with PCS solution and blocked for 1 hr at room temperature in PCS. Following block, cells were incubated with 1:100 dilution of rabbit anti-ovalbumin (OVA) antibody to stain OVA (Thermo/Pierce, Catalog # PA1-196) diluted in PCS solution overnight at 4°C. Washed cells were incubated with a 1:200 dilution of goat anti-rabbit Alexa Fluor 647 antibody (Thermo/Molecular Probes, Catalog # A21235) and 3.3 nM 488 phallodin (dissolved at 6.6 µM in methanol) in PCS solution in the dark for 60 min at room temperature. Wells were then washed $3 \times 5$ min in PCS solution. Cells were covered in 200 µl PBS and immediately imaged. Ovalbumin signal was quantified as the corrected total cell fluorescence (CTCF). CTCF = Integrated Density – Area of Selected Cell * Mean Fluorescence of 3 Background Readings. Each data point represents a single cell, and the graphs reflect pooled results from three biological replicates.

### Image processing and analysis

All image quantification was done on raw, unedited images. All images in figures were first analyzed in ImageJ, where a single Z-slice at the center of the cell was extracted. The image intensities were scaled to enhance contrast and cropped in Photoshop. For movies, background was subtracted in Fiji using a rolling ball radius of 50 µm and bleach corrected using the Histogram Matching plug in.

### Statistics

All statistical analysis was performed in Prism 6.0 (GraphPad, Inc.). The statistical test used is indicated in each figure legend. Error bars throughout the paper denote 95% confidence intervals of the mean. *** indicates $p<0.0001$; ** indicates $p<0.001$ and * indicates $p<0.01$.

## Acknowledgements

We thank J Reiter and E Yu for providing mouse long bones as a source for hematopoetic stem cells. We thank K McKinley and X Su for critical feedback on this manuscript. MAM was supported by the National Institute of General Medical Sciences of the National Institutes of Health under award number F32GM120990. APW was supported by a CRI Irvington Postdoctoral Fellowship. This work was funded by the Howard Hughes Medical Institute.

## Additional information

### Funding

| Funder | Grant reference number | Author |
|---|---|---|
| National Institute of General Medical Sciences | F32GM120990 | Meghan A Morrissey |
| Cancer Research Institute | Irvington Postdoctoral Fellowship | Adam P Williamson |
| Howard Hughes Medical Institute | | Ronald Vale |

The funders had no role in study design, data collection and interpretation, or the decision to submit the work for publication.

### Author contributions

Meghan A Morrissey, Adam P Williamson, Conceptualization, Formal analysis, Funding acquisition, Validation, Investigation, Visualization, Methodology, Writing—original draft, Writing—review and editing; Adriana M Steinbach, Conceptualization, Investigation, Visualization, Writing—review and editing; Edward W Roberts, Conceptualization, Resources, Formal analysis, Investigation, Methodology, Writing—review and editing; Nadja Kern, Formal analysis, Investigation, Visualization, Writing—review and editing; Mark B Headley, Conceptualization, Resources; Ronald D Vale, Conceptualization, Resources, Supervision, Funding acquisition, Writing—review and editing

Author ORCIDs

Meghan A Morrissey http://orcid.org/0000-0002-0531-4864

Adam P Williamson http://orcid.org/0000-0001-8905-5646

Ronald D Vale http://orcid.org/0000-0003-3460-2758

Ethics

Animal experimentation: All mice were maintained under specific pathogen-free conditions and treated in accordance with the regulatory standards of the NIH and American Association of Laboratory Animal Care standards, and are consistent with the UCSF Institution of Animal Care and Use Committee (IACUC approval: AN170208-01I).

Decision letter and Author response

Decision letter https://doi.org/10.7554/eLife.36688.021

Author response https://doi.org/10.7554/eLife.36688.022

# Additional files

## Supplementary files

• Transparent reporting form

DOI: https://doi.org/10.7554/eLife.36688.017

## Data availability

The replicates used to construct Figure 1d have been uploaded to Dryad (doi:10.5061/dryad.c57c1s0). Due to the large size of the datasets, the full set of raw images are available from the authors upon request.

The following dataset was generated:

| Author(s) | Year | Dataset title | Dataset URL | Database, license, and accessibility information |
| --- | --- | --- | --- | --- |
| Morrissey MA, Williamson AP, Steinbach AM, Roberts EW, Kern N, Headley MB, Vale RD | 2018 | Data from: Chimeric antigen receptors that trigger phagocytosis | https://doi.org/10.5061/dryad.c57c1s0 | Available at Dryad Digital Repository under a CC0 Public Domain Dedication |

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
