## [Decision Letter]

Thank you for submitting your article "Chimeric antigen receptors that trigger phagocytosis" for consideration by *eLife*. Your article has been reviewed by two peer reviewers, and the evaluation has been overseen by Jonathan Cooper as the Senior/Reviewing Editor. The following individuals involved in review of your submission have agreed to reveal their identity: Dennis Discher (Reviewer #1); Adam Hoppe (Reviewer #2).

The reviewers have discussed the reviews with one another and the Reviewing Editor has drafted this decision to help you prepare a revised submission.

Summary:

This manuscript describes engineering of chimeric receptors that when expressed in macrophages can enable target recognition of the CD19 or CD22 antigens and mediate phagocytosis and trogocytosis. The study works as a proof of principle showing a new effector function (phagocytosis) for chimeric antigen receptors in a new cell type (macrophages). Given the strong interest in cell-based immunotherapies such as chimeric antigen receptor-T cells this work will likely be of broad interest to the readership of *eLife*.

Essential revisions:

1) The work is timely as it suggests a connection between adoptive transfer therapies using engineered immune cells (such as CAR-T therapy) and immunotherapy approaches using macrophages (as cited studies using blockade of the CD47-SIRPalpha axis). However, whether such a connection can be realized remains unclear as only macrophage cell lines were engineered and phagocytosis assays were all conducted in vitro. A possible route forward is suggested by the engineering of CAR-P bone marrow-derived dendritic cells in Figure 4—figure supplement 4, but bone marrow-derived macrophages or other primary sourced MPS cells should be considered. It should be relatively easy for the authors to obtain such cell types, engineer them, and test them.

2) The phagocytic response to B-cell targets appears relatively weak (e.g. 2-6 macrophage having internalized B-cells) and seems to strongly favor a trogocytic response (nearly 100% of the macrophages having internalized B-cell membrane). Perhaps this is unique to the cell lines (macrophages and B-cells) used in the study, but it raises the question how of how similar would these results be to ADCP mediated by IgG2a (murine) or IgG1 (human) anti-CD19 antibodies under the same conditions? To this end, the authors should compare the CAR-P response to that stimulated by antibody-opsonized particles and/or cell. CD19-coated particles should be treated with anti-CD19 antibodies and incubated with cells for phagocytosis. While the specificity of CAR-Ps could be advantageous, true specificity for the CAR-P target is unlikely since Fc-receptor mediated pathways remain in place. For example, in the authors' system, CD22-coated particles would likely be phagocytosed by anti-CD19 CAR-P cells in the presence of an anti-CD22 antibody.

3) (Optional) The work would be more impactful if the authors demonstrated successful phagocytosis with different CAR extracellular domains that target antigens on solid tumors. As stated in the Introduction, macrophages might offer the benefit of trafficking into solid tumors better than T cells, yet the examples in this manuscript are the standard B cell targets CD19 and CD22. Adding such studies would be beneficial, but may be best left for a follow up paper.

---

## [Author Response]

Essential revisions:1) The work is timely as it suggests a connection between adoptive transfer therapies using engineered immune cells (such as CAR-T therapy) and immunotherapy approaches using macrophages (as cited studies using blockade of the CD47-SIRPalpha axis). However, whether such a connection can be realized remains unclear as only macrophage cell lines were engineered and phagocytosis assays were all conducted in vitro. A possible route forward is suggested by the engineering of CAR-P bone marrow-derived dendritic cells in Figure 4—figure supplement 4, but bone marrow-derived macrophages or other primary sourced MPS cells should be considered. It should be relatively easy for the authors to obtain such cell types, engineer them, and test them.

We agree that expressing CAR-P in primary cells such as Bone Marrow Derived Macrophages is an essential step towards a possible therapeutic option. At the suggestion of the reviewers, we expressed the CAR-P^FcRɣ^ in murine Bone Marrow Derived Macrophages (BMDMs). The BMDMs robustly engulfed CD19 beads and these data are now included in Figure 1. We feel that the discovery that CAR-P is transferrable to primary cells is a key new addition to the paper and we thank the reviewers for their suggestion to pursue this experiment.

2) The phagocytic response to B-cell targets appears relatively weak (e.g. 2-6 macrophage having internalized B-cells) and seems to strongly favor a trogocytic response (nearly 100% of the macrophages having internalized B-cell membrane). Perhaps this is unique to the cell lines (macrophages and B-cells) used in the study, but it raises the question how of how similar would these results be to ADCP mediated by IgG2a (murine) or IgG1 (human) anti-CD19 antibodies under the same conditions? To this end, the authors should compare the CAR-P response to that stimulated by antibody-opsonized particles and/or cell. CD19-coated particles should be treated with anti-CD19 antibodies and incubated with cells for phagocytosis. While the specificity of CAR-Ps could be advantageous, true specificity for the CAR-P target is unlikely since Fc-receptor mediated pathways remain in place. For example, in the authors' system, CD22-coated particles would likely be phagocytosed by anti-CD19 CAR-P cells in the presence of an anti-CD22 antibody.

We also agree that this is a key question, and have now included data to address this point. When we performed the bead size analysis presented in Figure 2B, we also included 10 µm beads coated in phosphatidylserine, the ligand for the endogenous Megf10, as a positive control. The phosphatidylserine beads also contained an integrin ligand, ICAM-1, to the beads to promote adhesion. We found that CAR-P macrophages internalized CD19-coated beads and phosphatidylserine beads to a similar degree. This data has been added to Figure 2B.

To address the reviewers’ question about whether opsonizing whole Raji cells with anti-CD19 antibodies would enhance whole cell eating we added a mouse anti-human CD19 IgG2a at a final concentration of 20 µg/ml. We found that this did not enhance whole cell internalization, indicating that the CAR-P system is not less efficient than the endogenous IgG system (Figure 4—figure supplement 3).

3) (Optional) The work would be more impactful if the authors demonstrated successful phagocytosis with different CAR extracellular domains that target antigens on solid tumors. As stated in the Introduction, macrophages might offer the benefit of trafficking into solid tumors better than T cells, yet the examples in this manuscript are the standard B cell targets CD19 and CD22. Adding such studies would be beneficial, but may be best left for a follow up paper.

We agree that targeting CAR-P towards solid tumors is an exciting future avenue of research. However, because of the time required to develop and characterize new CARs and assay platforms we consider these experiments beyond the scope of the current work.